# Using sero-epidemiology to monitor disparities in vaccination and infection with SARS-CoV-2

Isobel Routledge [1,6 ✉], Saki Takahashi [1,6 ✉], Adrienne Epstein [1], Jill Hakim[1], Owen Janson[1], Keirstinne Turcios[1], Jo Vinden[1,2], John Tomas Risos[3], Margaret Rose Baniqued[3], Lori Pham[3], Clara Di Germanio[4], Michael Busch [3,4], Margot Kushel[5], Bryan Greenhouse [1] & Isabel Rodríguez-Barraquer[1 ✉]

As SARS-CoV-2 continues to spread and vaccines are rolled-out, the "double burden" of disparities in exposure and vaccination intersect to determine patterns of infection, immunity, and mortality. Serology provides a unique opportunity to measure prior infection and vaccination simultaneously. Leveraging algorithmically-selected residual sera from two hospital networks in the city of San Francisco, cross-sectional samples from 1,014 individuals from February 4–17, 2021 were each tested on two assays (Ortho Clinical Diagnostics VITROS Anti-SARS-CoV-2 and Roche Elecsys Anti-SARS-CoV-2), capturing the first year of the epidemic and early roll-out of vaccination. We estimated, using Bayesian estimation of infection and vaccination, that infection risk of Hispanic/Latinx residents was five times greater than of White residents aged 18–64 (95% Credible Interval (CrI): 3.2–10.3), and that White residents over 65 were twice as likely to be vaccinated as Black/African American residents (95% CrI: 1.1–4.6). We found that socioeconomically-deprived zipcodes had higher infection probabilities and lower vaccination coverage than wealthier zipcodes. While vaccination has created a 'light at the end of the tunnel' for this pandemic, ongoing challenges in achieving and maintaining equity must also be considered.

[1] Department of Medicine, University of California San Francisco, San Francisco, CA, USA. [2] Infectious Disease and Immunity Graduate Group, University of California Berkeley, Berkeley, CA, USA. [3] Department of Laboratory Medicine, University of California San Francisco, San Francisco, CA, USA. [4] Vitalant Research Institute, San Francisco, CA, USA. [5] Center for Vulnerable Populations, Zuckerberg San Francisco General Hospital and Trauma Center, University of California San Francisco, San Francisco, CA, USA. [6] These authors contributed equally: Isobel Routledge, Saki Takahashi. ✉email: isobel.routledge@ucsf.edu; saki.takahashi@ucsf.edu; Isabel.rodriguez@ucsf.edu

During the initial waves of the COVID-19 pandemic, disparities in disease burden were largely driven by differences in infection rates[1,2]. In addition, structural inequalities are associated with disparities in the risk of comorbidities (which increase the likelihood of hospitalization and death from COVID-19[3]) and with disparities in access to healthcare (both in managing comorbidities and in accessing care for COVID-19). As vaccine and booster roll-outs continue to advance in the United States and globally, disparities also exist in both vaccine access and uptake. These disparities are multifactorial and complex, including reduced technology access and literacy[4], barriers in access to healthcare, concern about the safety of the vaccines[5,6], mistrust as a result of a history of medical racism and discrimination[6–9], and poor access to reliable information about the vaccine[6,10]. In the age of vaccination, policymakers must understand the way in which societal structures affect disparities in both infection and vaccination. These disparities may interact to affect both population level immunity and the burden of COVID-19 in different communities. This is relevant both in the present and in the future, as policymakers consider the continued roll-out of vaccines in the context of new variants, as well as preparing for and responding to other diseases.

Given the high levels of disease under-ascertainment, serology (i.e., the measurement of antibodies) has been particularly useful for understanding SARS-CoV-2 infection levels in the population. When there is variability in testing rates and healthcare seeking behavior, serology is an even more useful tool. Seroepidemiology provides a unique opportunity to measure biomarkers of infection and vaccination simultaneously, and to relate these metrics to demographic and geographic factors. In settings where vaccines based on the SARS-CoV-2 spike protein (e.g., currently available mRNA or adenovirus vector vaccines) are used, measuring long-lived antibody responses to both spike and non-spike proteins can be used to disentangle immune responses elicited by infection from vaccination. While structural inequalities are by no means limited to the United States, here we focus on a case example leveraging serology to understand inequalities within this context.

San Francisco is a city which has had a relatively successful early and sustained COVID-19 response and has achieved high vaccination coverage. However, reported case counts of COVID-19 and hospitalization rates have been higher in socioeconomically deprived areas, amongst homeless individuals, and within the city's Hispanic/Latinx and Black/African American communities[11,12]. Disparities in vaccination coverage have also been reported, particularly in the early months of vaccine roll-out, creating a double burden for some vulnerable communities. In San Francisco, whilst some disparities have now been addressed, vaccination remains much lower in homeless individuals and in Black/African American individuals[13].

To measure disparities in both infection rates and vaccination, we leveraged a SARS-CoV-2 serosurveillance platform launched in March 2020 that utilizes residual blood samples taken from two hospital networks in San Francisco. Estimates derived from this platform during the first wave of the pandemic showed seroprevalence in Hispanic/Latinx individuals to be nearly two times higher than in White individuals, and nearly two times higher in homeless individuals than the population average[14]. We collected samples from individuals undergoing routine blood draws between February 4 and February 17, 2021, capturing transmission during the first 11 months of the epidemic and the early roll-out of vaccination (primarily for those over 65 years old as well as workers in specific professions i.e., healthcare). These samples were tested using two serologic assays: one detecting antibodies to SARS-CoV-2 elicited by infection and not by vaccines currently used in the US, and one detecting antibodies to SARS-CoV-2 elicited by both infection and vaccination. We used Bayesian statistical models to estimate the proportion of the population that was seropositive due to natural infection and the proportion seropositive due to vaccination, stratified by age, race and zipcode of residence.

## Results

Between February 4, 2021, and February 17, 2021, we collected samples from 1014 individual patients, from UCSF Health ($n = 698$ patients) and the San Francisco Department of Public Health ($n = 316$ patients) networks. By design, the geographic distribution of residents matched the proportion of the San Francisco population living in each zipcode (Fig. 1). Our sample was equally distributed by sex; however, our sample over-represented the 65+ age range and under-represented the 0-34 age range relative to the San Francisco population (Table 1). Our results were relatively representative of the San Francisco population by race and ethnicity, although our sample over-represented those who identified as Black/African American and under-represented those who identified as Asian.

Following testing samples on both Vitros (spike protein) and Roche (nucleocapsid protein) assays, of the sampled population where assay results were complete ($N = 915$), we found that while 28.4% ($N = 260$) tested positive on the Vitros assay and therefore antibodies to SARS-CoV-2 were detected, only 8.6% ($N = 79$) tested positive on the Roche assay, detecting antibodies elicited by prior natural infection (i.e., natural infection prior to serosurvey) (Fig. 2, Supplementary Fig. 1, Supplementary Table 1). Of the 999 samples where assay results were available, $N = 81$ samples were excluded from the bivariate analyses due to missing data in at least one assay, and 3 additional samples were excluded due to positive results on the Roche assay despite negative results on the Vitros assay.

Our estimated probabilities of vaccination and prior infection stratified by age, race/ethnicity and zipcode showed striking differences in prior infection rates and vaccination rates across the city (Fig. 3, Supplementary Fig. 2, Supplementary Table 2). Zipcodes in the southeastern region of the city, comprising medically underserved neighborhoods, had demonstrably higher rates of prior infection and lower rates of vaccination. This pattern is not evident in estimated seroprevalence by the Vitros assay, which captures antibody responses acquired through natural infection and/or vaccination. For example, within the 94124 zipcode, Bayview-Hunter's Point, one of the city's most deprived zipcodes, the mean probability of prior infection was 0.155 (95% credible interval (CrI): 0.077–0.254) and vaccination was 0.079 (95% CrI: 0.019–0.163), whereas 94115, Pacific Heights, one of the wealthier zipcodes in San Francisco with a median household income almost double that of Bayview-Hunter's Point at $123,037[15], the probability of infection was just 0.023 (95% CrI: 0.001–0.080) and vaccination was 0.359 (95% CrI: 0.258–0.467). Supplementary Fig. 3 maps the socioeconomic and racial/ethnic make-up of zipcodes within San Francisco to further contextualise Fig. 3.

We found the highest seroprevalence as a result of prior infection in younger age groups (using the Roche assay, Fig. 4a, Supplementary Table 3). We estimated seroprevalence derived from both vaccination and natural infection using the Vitros assay to be much higher in those aged over 65 (Fig. 4b, Supplementary Table 3), consistent with the eligibility criteria for vaccination in the weeks before the sampling period.

We identified differences in prior infection rates by race/ethnicity (Fig. 4c, d): we estimated that the risk of prior infection of Hispanic/Latinx residents was 5.3 (95% CrI: 3.2–10.3) times greater than the risk of White residents aged 18–64 (Fig. 5a, Supplementary Table 4).

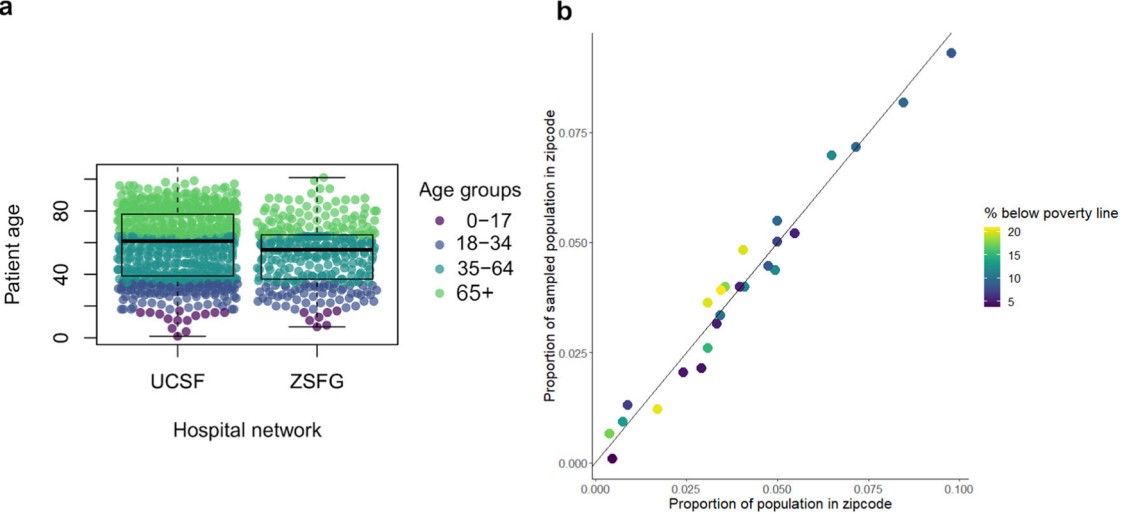

**Fig. 1 Sample characteristics. a** Age distribution by health network of sample collection within the University of California, San Francisco (UCSF) (N = 698 biologically independent samples) and San Francisco Department of Public Health (ZSFG) hospital networks (N = 316 biologically independent samples). Each point represents a sample and colors correspond to age bins used for analysis. The box plot whiskers show the maximum and minimum ages, the box shows the 25-75[th] percentile values for age, and the central line within the boxplot represents the median age within each health network. **b** Proportion of samples from a given San Francisco zipcode plotted against the proportion of the San Francisco population within that zipcode. Colors show the percentage of residents below the poverty line within that zipcode, as determined by the American Community Survey 2019[15], using census bureau definitions of poverty thresholds[35].

| Table 1 Sample characteristics. | | | |
|---|---|---|---|
| | **N** | **%** | **SF Population (ACS 2019)** |
| Age | | | |
| 0–17 | 21 | 2.10% | 13.40% |
| 18–34 | 157 | 15.50% | 30.60% |
| 35–64 | 442 | 43.60% | 40.60% |
| 65+ | 393 | 38.80% | 15.40% |
| Unknown | 1 | 0.10% | N/A |
| Sex | | | |
| Female | 509 | 50.20% | 49.30% |
| Male | 504 | 49.70% | 50.70% |
| Unknown | 1 | 0.10% | N/A |
| Hospital | | | |
| University of California San Francisco (UCSF) Health | 698 | 68.80% | N/A |
| Zuckerberg San Francisco General Hospital (ZSFG)/San Francisco Department of Public Health (SFDPH) | 316 | 31.20% | N/A |
| Race/Ethnicity | | | |
| Asian | 282 | 27.80% | 34.60% |
| Black/African American | 115 | 11.30% | 5.20% |
| Hispanic/Latinx | 175 | 17.30% | 15.20% |
| Other | 75 | 7.40% | 5.20% |
| White | 367 | 36.20% | 39.80% |

Table showing the sample size and distribution of the sample population by demography and hospital system, compared to the San Francisco Population as determined by the American Community Survey 2019[15].

These trends were echoed in older individuals (aged 65 + ) (Fig. 5a, Supplementary Table 4). We also identified disparities in vaccination coverage among the 65+ year old population, who were eligible to receive the vaccine during this time period. We estimated that White San Francisco residents over the age of 65 were twice as likely (2.0, 95% CrI: 1.1–4.6) to be vaccinated as Black/African American residents.

Taken together, these findings imply that there is an imbalance between the risk of infection and the early rate of vaccination in certain populations. Among the 65+ year old population, we found greatly increased ratios of vaccination coverage as compared to infection risk among Asian and White individuals, while these ratios were much lower among Black/African American and Hispanic/Latinx individuals (Fig. 5b, Supplementary Table 4). For every naturally infected Asian resident of this age group, there were 12.2 vaccinated Asian residents (95% CrI: 4.2–55.5), whereas, for every naturally infected Hispanic/Latinx resident of this age group, there were only 1.6 vaccinated Hispanic/Latinx residents (95% CrI: 0.7–3.7). For both Hispanic/Latinx and Black/African American individuals over 65 years old, the risk of having immunity acquired through vaccination, relative to natural infection, was up to four times lower than for White individuals.

## Discussion

Using data from a previously established serosurveillance platform for SARS-CoV-2 in San Francisco, we quantified disparities in both vaccination coverage and infection rates across different demographic groups and geographies, and show that, during early vaccine roll-out, vaccination coverage was much higher in Asian and White populations, despite experiencing lower risk of infection by SARS-CoV-2 than Black/African American and Hispanic/Latinx populations.

The "double burden" we observed in San Francisco during the early vaccine roll-out echoes broader patterns that have been observed in San Francisco and elsewhere. Even though San Francisco was hailed as the first major US city to reach the milestone of 80% vaccination coverage in adults[16], recent increases in infection have been found to be concentrated in the neighborhoods which were hardest hit by initial infections and where we found vaccination-related immunity was lowest[17]. A report from the University of Texas found striking geographic and racial stratification of cases of COVID-19 and vaccination rates in Austin, Texas, which also closely mapped with indices of deprivation and social

vulnerability over zipcodes[18]. Like in San Francisco, the neighborhoods which were predominantly Hispanic/Latinx communities and had higher indices of deprivation also had higher incidence of SARS-CoV-2 infection and lower vaccination coverage. Disparities in SARS-CoV-2 vaccination coverage among socially vulnerable populations have been documented across the United States[19] and in other parts of the world[20].

There are several caveats and limitations to the approach introduced here. The Roche assay can only differentiate antibody responses resulting from natural infections in settings where Spike-based vaccines are used (that do not generate antibody responses against the nucleocapsid), so in geographies where other vaccines are used, this approach would not be suitable. Although we used samples obtained through the University of California San Francisco and San Francisco Department of Public Health (SFDPH) health networks, which allowed for the inclusion of un-insured and under-insured individuals, we still are only able to capture those seeking healthcare or are reached by the SFDPH. Although the exclusion of non-emergency room inpatients and those seeking care for COVID-19 symptoms and inclusion of only routinely collected blood tests all were designed to reduce the biases in our sample, this study still may not capture a fully representative cross-section of the population. As eligibility in San Francisco for vaccination among individuals under 65 years of age during our sampling period was limited to healthcare workers, we necessarily aggregated vaccination coverage among 18–64 year olds into a single age group (Supplementary Fig. 4). In addition, given the short duration between the start of vaccine roll-out in San Francisco and this serosurvey, these data are only able to capture vaccination coverage and inequities therein during the early roll-out.

While inequalities revealed during COVID-19 are not new, the pandemic has highlighted the ways in which even a city such as San Francisco which invests deeply in public health and social safety nets[21] still has deep structural inequalities. These have been proposed to be the result of a combination of higher infection rates[11], incompatibility of living or work conditions with risk reduction[8,11], and lower or delayed access to and uptake of vaccines as they were initially being rolled out[22].

Various initiatives are underway around the country to pinpoint geographic and other disparities in the context of COVID-19[23,24]. However, as well as highlighting disparities, it is important to consider what testing and vaccination initiatives have been successful to reduce these inequities. For example, within San Francisco, robust community-academic partnerships have been key for effectively responding to the pandemic in vulnerable communities[25,26] and for narrowing gaps in vaccination coverage, such as through low-barrier neighborhood vaccination sites[27]. Prospectively, as we continue to gain a better understanding of waning immunity and continue to roll-out vaccine booster doses, considerations of equity will remain an important consideration for allocating resources. In the context of the United States, where vaccination and infection elicit different immune responses, serology provides a powerful lens through which we can quantify these disparities directly[28]. In addition, it is important to consider the desired metric before conducting a serosurvey, as assays measure different pathways to immunity and any disparities in infection rates may be masked by using assays that measure overall antibody prevalence.

Since the early days of the pandemic, many policy recommendations have been made for ways to reduce health disparities in infection[29] and vaccination[30]. Policymakers must invest in addressing both the upstream, structural drivers of health disparities, such as providing workers with a living wage, affordable housing, and access to quality healthcare, and also downstream drivers such as improved community engagement, targeted testing and vaccination provision, and assistance with common barriers to accessing healthcare such as technology access/literacy,

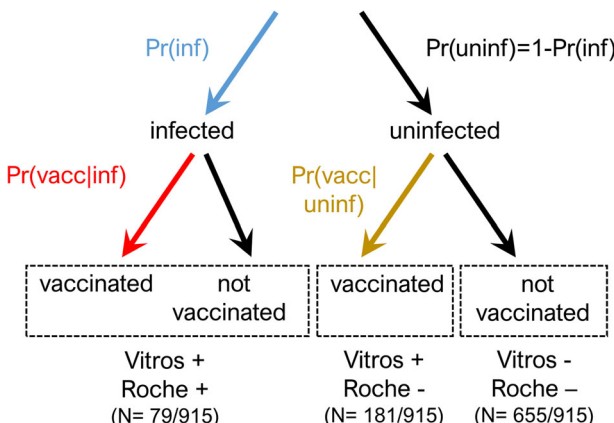

**Fig. 2 Schematic of parameters to be estimated using serosurveillance platform (shown in red, blue and gold).** Red represents the probability of vaccination given prior infection, Pr(vacc|inf), blue represents the probability of prior infection, Pr(inf), and gold represents the probability of vaccination given no prior history of infection, Pr(vacc|uninf).

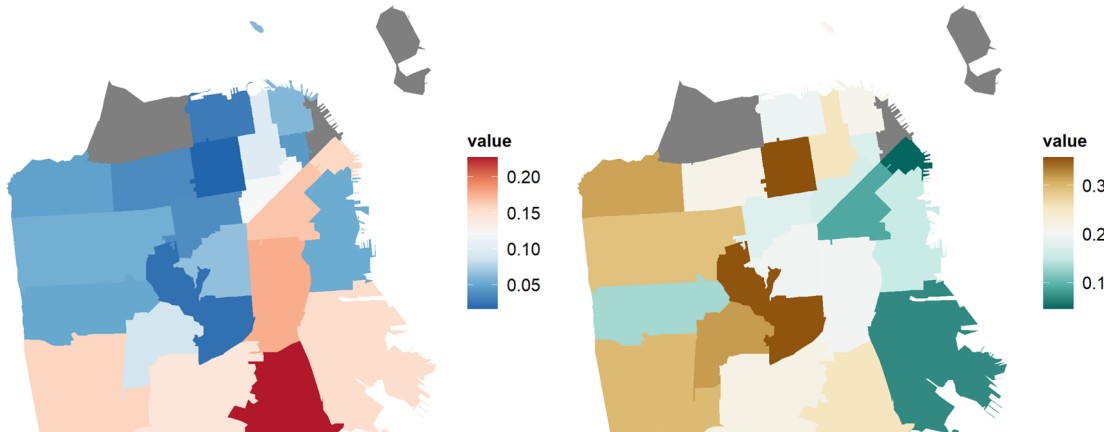

**Fig. 3 Maps showing geographic disparities in SARS-CoV-2 within San Francisco.** Maps show **a** estimated probability of prior infection and **b** probability of vaccination by zipcode in San Francisco, as of February 2021.

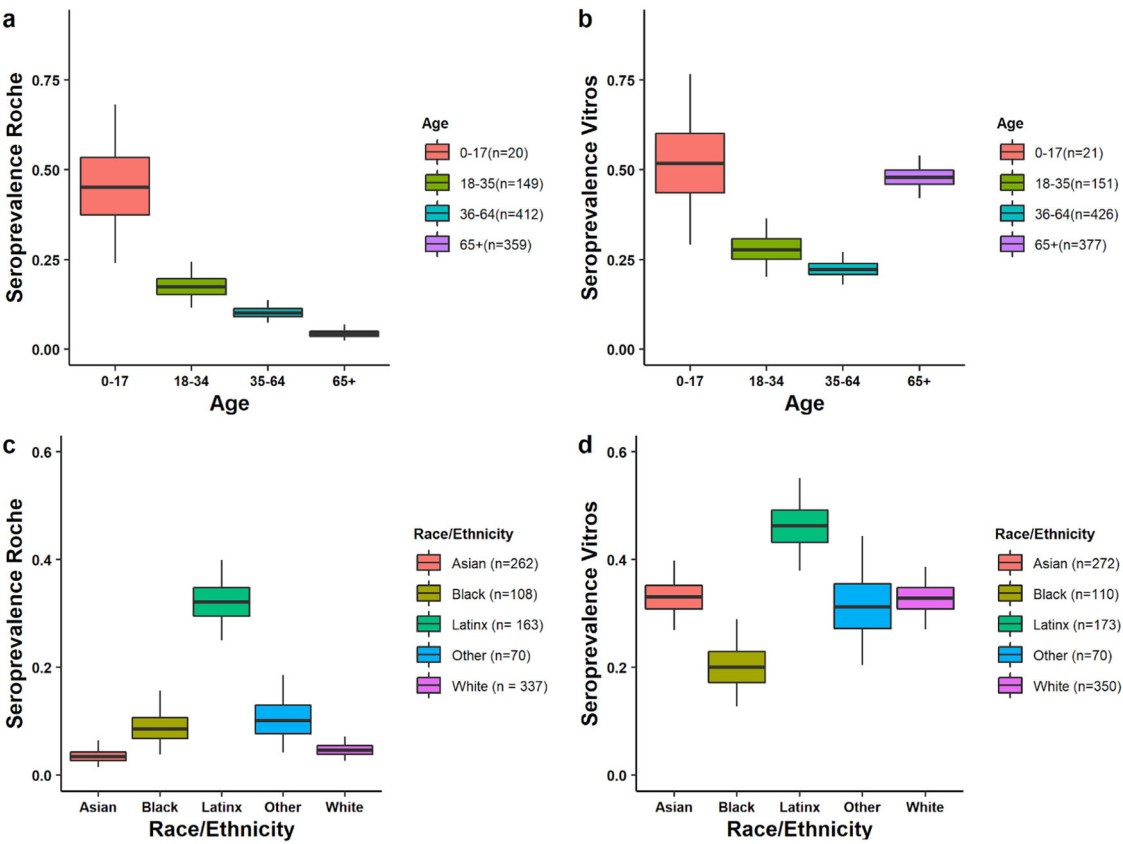

**Fig. 4 Stratified seroprevalence by assay and by demographic group. a** Univariate Roche seropositivity estimates by age (elicited by prior infection). **b** Univariate Vitros estimates by age (elicited by either prior infection or vaccination). **c** Univariate Roche estimates by race/ethnicity. **d** Univariate Vitros estimates by race/ethnicity. For all panels, the center line of the box and whisker diagram represents the median posterior estimate, the box represents the 25-75th percentile values of the posterior, and the whisker lines show the 95% Credible Interval.

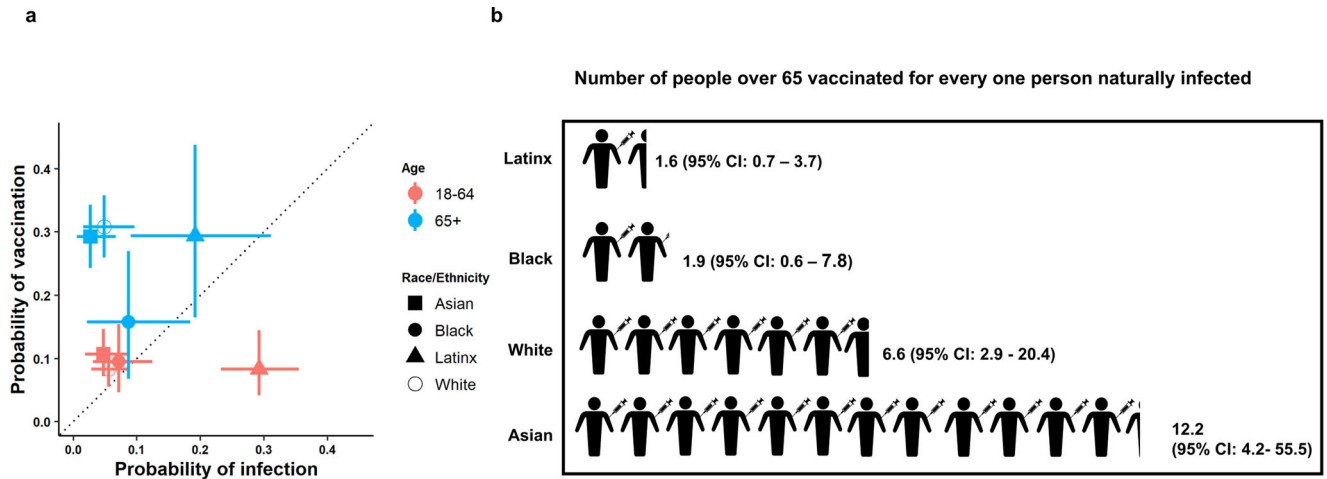

**Fig. 5 Relationship between probability of vaccination and probability of prior infection by race/ethnicity. a** Probability of infection vs. probability of vaccination by age and race/ethnicity. Error bars show the upper and lower limits of the 95% Credible Interval of the estimates, and points show the median of the estimates. (65+ and Black/African American, $N = 33$; 65+ and White, $N = 160$; 65+ and Hispanic/Latinx, $N = 37$; 18–64 and Black/African American, $N = 82$; 18–64 and White, $N = 202$; 18–64 and Hispanic/Latinx, $N = 130$). **b** Infographic showing the number of estimated people vaccinated for every one person previously naturally infected in San Francisco within each racial/demographic group.

transport, and providing accessible health information in multiple languages. While the arrival of the SARS-CoV-2 vaccine has created a 'light at the end of the tunnel' for this pandemic, ongoing challenges that long predate COVID-19 in achieving and maintaining equity must also be considered.

## Methods

As part of an existing serological survey[14], residual serum samples from routine blood draws from the University of California, San Francisco (UCSF) and San Francisco Department of Public Health (SFDPH) inpatient and outpatient healthcare systems were sampled between February 4 and February 17, 2021 (Supplementary Fig. 5). UCSF Medical Center is a network of three hospitals with

~1.8 million outpatient visits annually (https://www.ucsfhealth.org/about/annual-reports). The SFDPH hospital, Zuckerberg San Francisco General Hospital (ZSFG), is a city hospital that provides trauma, medical, and surgical services to a heterogeneous population of largely un- or underinsured patients, including the city's homeless population, and serves roughly 100,000 patients per year (https://zsfg.ucsf.edu/about-ucsf-zsfg). A total of 1,091 samples were collected, of which 77 were excluded due to participation in a separate COVID research study, and a further 15 were later excluded from further analyses as they could not be linked to antibody test results. The characteristics of the samples are illustrated in Table 1. The full inclusion and exclusion criteria and sampling algorithm are described previously[14]. Briefly, we included patients undergoing routine blood testing, defined as blood chemistries and tests for sexually transmitted infections and rubella. We included patients residing in San Francisco, including those experiencing homelessness. We excluded individuals who were tested for SARS-CoV-2 during the visit when they received their blood draw (except if the test was for routine purposes, such as testing prior to an elective procedure or admittance to the hospital). We did not have any exclusion criteria for previous visits or tests for SARS-CoV-2 of any severity. We restricted our sample to outpatient and emergency department visits for adults; for under 18 s, we included both inpatient and outpatient visits due to small numbers of available samples. Finally, we excluded samples if a sample from the same patient had been selected within the previous 14 days. All demographic data were obtained from the electronic health record data. For race/ethnicity categorisation we aggregated the clinical records for self-reported race (Options: Black/African American, White, Asian, Hawaiian/Pacific Islander, Native American, Other, Do not wish to disclose) and self-reported ethnicity (Hispanic/Latinx, Not Hispanic/Latinx, Do not wish to disclose), so that those who self identify as Hispanic/Latinx of any race were categorised as Hispanic/Latinx for this study and all others were categorised by their self-reported race.

Each sample ($N = 1,014$) was tested on two commercial SARS-CoV-2 serologic platforms. The Ortho Clinical Diagnostics VITROS Anti-SARS-CoV-2 Total assay measures the total Ig antibody response to the S1 subunit of the SARS-CoV-2 spike (S) protein and therefore is expected to yield a positive results after natural infection or vaccination[31]. The Roche Elecsys Anti-SARS-CoV-2 assay measures the total Ig antibody response to the SARS-CoV-2 nucleocapsid (N) protein[32] and therefore is expected to yield a positive results after natural infection, but not after vaccination with vaccines based on the spike protein. In a previous analysis assessing the test performance characteristics of many SARS-CoV-2 serological assays, we found both assays to exhibit high sensitivity over time following infection[33].

Seropositivity on the Vitros assay indicates whether or not an individual has had any prior immune response to SARS-CoV-2, either through natural infection and/or vaccination. The SARS-CoV-2 mRNA and adenovirus vector vaccines elicit immune responses to only the S protein of the virus. Therefore, in contexts where these vaccines are used exclusively (such as the United States), seropositivity on the Roche assay indicates whether an individual has had a prior immune response to SARS-CoV-2 via infection. Assuming perfect test performance characteristics, the difference between the proportion seropositive on Vitros and the proportion seropositive on Roche indicates the proportion of the population that has been vaccinated and has not been infected.

We used Binomial models in a Bayesian framework to first estimate seropositivity separately by assay. We adjusted for the manufacturer-reported specificity of each assay (100% for Vitros and 99.80% for Roche) and for in-house estimates of the sensitivity of each assay (at 2 months post symptom onset among non-hospitalized individuals based on a longitudinal post-infection study[33]), corresponding to 83.8% for Vitros and 90.0% for Roche. We note that sensitivity was particularly consistent over time following infection for these assays, so our results are robust to the choice of exact time point used. By using the manufacturer-reported specificity values to adjust estimates, we assume that they are unbiased and reflect specificity in the general population. We computed 95% credible intervals (CrI) to quantify uncertainty in posterior estimates. For these univariate analyses, age was stratified into 4 groups (0–17 y, 18–34 y, 35–64 y, and 65 + y) and race/ethnicity was stratified into 5 groups (Asian, Black/African American, Hispanic/Latinx, White, and Other).

We then conducted bivariate analyses using the results of each sample on both assays (see Fig. 2 for a schematic). We used the data set of individuals who had a test result on both the Vitros and Roche assays, and removed the 81 samples that had a result on only one assay. In addition, we removed the 3 samples that tested negative on Vitros and positive on Roche, which likely reflects a false negative result on the Vitros assay and/or a false positive result on the Roche assay (Supplementary Table 4).

For demographic analyses, age was stratified into 2 groups (18-64 y and 65 + y), and race/ethnicity was stratified into 4 groups (Asian, Black/African American, Hispanic/Latinx, and White). We omitted individuals aged 0–17 y from this portion of the analysis, as individuals in that age group were not eligible for vaccination during this time frame in San Francisco (February 2021). The decision to choose 65 years as a cutoff was due to the age-based roll-out of SARS-CoV-2 vaccination and the expected resultant differences in vaccine coverage by age during the time period of this serosurvey. We also omitted individuals with race/ethnicity of "Other" due to lack of data on vaccine doses in this demographic group that are used as a prior for estimation (see below). For each combination of age group and race/ethnicity j, we estimated the marginal probabilities of natural

infection, Pr(inf), and of vaccination, Pr(vacc), separately as follows:

$$N_{\text{Vitros+ve&Roche+ve},j} \sim \text{Binomial}(N_j, \text{Pr(inf)}) \quad (1)$$

$$N_{\text{Vitros+ve&Roche-ve},j} \sim \text{Binomial}(N_j, \text{Pr(vacc|uninf)} * [1 - \text{Pr(inf)}]) \quad (2)$$

$$N_{\text{Vitros-ve&Roche-ve},j} \sim \text{Binomial}(N_j, [1 - \text{Pr(vacc, |, uninf)}] * [1 - \text{Pr(inf)}]) \quad (3)$$

$$\text{Pr(vacc)} \sim \text{Beta}(\mu_j, \kappa_j) \quad (4)$$

$$\text{Pr(vacc)} \sim \text{Beta}(\mu_j, \kappa_j) \quad (5)$$

$N_j$ represents the number of individuals in our data set in group $j$ who were included, and $n_{x,j}$ represents the number of individuals in group $j$ with antibody results $x$. For this analysis, we assumed perfect test performance characteristics. As Pr(vacc|inf) is not identified by our data, and we do not assume that Pr(vacc|inf) = Pr(vacc|uninf), we set up a process to estimate the hyper-priors $\mu_j$ (mean) and $\kappa_j$ (precision) for each group $j$ based on reported vaccination coverage data. To do this, we first obtained the total population size $M_j$ of group $j$ in San Francisco as well as the reported number of individuals in that group $m_j$ who had been vaccinated up to January 20, 2021 (i.e., 3 weeks before the weighted mean date of sample collection, allowing for time to sero-conversion after vaccination). We then used a hypergeometric distribution to sample $N_j$ individuals without replacement from a population in which $m_j$ individuals had been vaccinated and $M_j$ - $m_j$ had not. We then calculated the empirical proportion of vaccinated individuals among the $N_j$ in that simulation, and repeated this procedure 10,000 times to obtain a prior distribution of Pr(vacc). This distribution was used to estimate the hyper-priors of the Beta distribution. This procedure was performed separately for each group $j$.

For geographic analyses, zipcodes with fewer than 10 individuals were excluded. As data on vaccine doses distributed by zipcode during this time-frame was not available, we modified the model above by assuming that vaccination was independent of infection, and estimated a single Pr(inf) and Pr(vacc) for each zipcode.

All Bayesian analyses were conducted in Stan version 2.21.2. Four chains with 2000 iterations were run, including a burn-in period of 1000 iterations. Convergence was confirmed using the R-hat statistic. All parameters where priors are not specifically described above had uninformative priors and were informed solely by the observed data.

This study received expedited review approval by the UCSF IRB #20-30379 (Serological Surveillance of SARS-CoV-2 in Residual Serum/Plasma Samples). The IRB did not require patient contact or written consent to use residual sera.

**Reporting summary**. Further information on research design is available in the Nature Research Reporting Summary linked to this article.

## Data availability
Data to reproduce Fig. 1a cannot be shared on an individual level due to institutional data privacy requirements but sample distributions by age band are shown in Table 1. Data used in Fig. 1b is shown in Supplementary Table 2, and from the 2019 American Community Survey Five Year Estimates[15]. Data to reproduce Fig. 2 is available in Supplementary Table 1. Data to reproduce Fig. 3 is in Supplementary Table 2. Data to reproduce Fig. 4 is in Supplementary Table 3. Data to reproduce Fig. 5 is in Supplementary Table 4. For enquiries to access individual data, please contact Isabel Rodriguez Barraquer Isabel.rodriguez@ucsf.edu and we will endeavour to respond within two weeks. Institutional policy requires the use of a data sharing contract for sharing clinical data.

## Code availability
All analysis was conducted using the R statistical software, version 4.0.3 and the Stan programming language. All code to reproduce these results are available at: https://github.com/EPPIcenter/scale-it-2[34] and the previously published algorithm used to select samples is available at https://github.com/EPPIcenter/scale-it.

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

## Acknowledgements

We acknowledge sources of funding support, including from the Schmidt Science Fellows, in partnership with the Rhodes Trust (S.T.); Chan Zuckerberg Biohub Investigator program (B.G.); the ZSFG Department of Medicine and Division of HIV, ID, and Global Medicine; the MIDAS Coordination Center COVID-19 Urgent Grant Program (MIDASNI2020-5) by a grant from the National Institute of General Medical Science (3U24GM132013-02S2) (I.R., S.T., I.R.B.); and the National Institutes of Health/National Institute of General Medical Sciences R35GM138361-02 (I.R.B.). We acknowledge Dr. Carina Marquez for helpful comments on the manuscript. We acknowledge Valerie Green and Phillip Williamson at Creative Testing Solutions for performing the Roche testing. We acknowledge the groups of Dr. Kara Lynch and Dr. Alan Wu for facilitating the collection of samples at ZSFG. We acknowledge the groups of Dr. Lee Besana and Dr. Marcelina Coh for facilitating the collection of samples at UCSF.

## Author contributions

I.R, S.T., A.E., and I.R.B. conceived of study; I.R. and S.T. carried out analysis; I.R., S.T., and A.E. wrote code/sampling algorithm; M.K., I.R.B., and B.G. provided supervision and guidance. J.H., O.J., K.T., J.V., J.T.R., M.R.B., L.P., C.D.G., and M.B. contributed to data acquisition and sample processing. I.R. and S.T. wrote paper and generated figures. All authors reviewed manuscript and provided comments.

## Competing interests

The authors declare no competing interests.
