## [Peer Review File · Nature Communications]

Using sero-epidemiology to monitor disparities in vaccination and infection with SARS-CoV-2REVIEWER COMMENTS

Reviewer #1 (Remarks to the Author):

The authors describe seroepidemiologic study results of a convenience sample with about one thousand blood specimens from inpatients and outpatients collected at the University of California San Francisco Medical Center and the San Francisco Department of Public Health between February 4 and February 17, 2021, during the early stages of COVID-19 vaccination efforts. Serologic testing results for COVID-19 spike and nucleocapsid protein antibody were analyzed using binomial Bayesian modeling. The raw natural infection rate as measured by nucleocapsid antibody was 8.6% and spike protein IgG antibody measuring both natural and vaccine induced antibody was 28.4%. Model adjusted natural infection rates were highest in Hispanic and black residents, while the highest estimated vaccination rates were in Asian and white San Francisco study populations. Patients 65+ years had lower natural infection rates and as expected higher estimated vaccination rates given vaccination eligibility during that time period. Lower income medically underserved zip codes in San Francisco also had higher infection and lower estimated vaccination rates. The authors discuss the challenges and public health measures needed to address these sociodemographic disparities in COVID-19 risk and prevention.

This paper is well written and uses appropriate and strong laboratory and biostatistical methodology despite the study design weakness of patients' blood specimens being a convenience sample (the authors address the relative representativeness of the study sample). Similar results regarding lower vaccination rates in areas with vulnerable populations have been seen in other parts of the United States but this rigorous seroepidemiologic analysis adds to the literature on the pandemic epidemiology in the U.S.

Specific comments:

- 1) Results, line 97 – The authors indicate that zip codes in southeastern San Francisco had the pattern of higher natural infection and lower vaccination rates across all age and demographic groups and reference Supplementary Table 2. Supplementary Table 2 does not stratify the zip code results by age and demographic groups. The low vaccination/high natural infection finding in these zip codes beg the question of whether these rates are related to the race/ethnic makeup of the zip codes or is an SES related phenomenon, or not an issue possible to disentangle.
- 2) Discussion paragraph, lines 175-179 – This one sentence paragraph is somewhat unclearly expressed. Structural inequalities in a wealthy progressive city such as San Francisco undoubtedly contribute to high infection rates and less vaccination in subpopulations. But does evidence exist for vaccine hesitancy, especially among the black population in San Francisco?
- 3) Methods, lines 256-247 – The authors adjusted for manufacturer reported specificity for each serology assay. Manufacturer reported specificities are often overstated based on the set of samples used to assess specificity. The authors should perhaps comment on how lower test specificities would affect the uncertainty in the estimated rates of positivity.
- 4) Discussion, paragraph lines 181-195 – The authors reference a preprint describing an intervention in a predominantly Latinx community in San Francisco. Were there similar projects for the black population in San Francisco?
- 5) References – The authors should probably reference Jones J et al, Estimated US Infection- and Vaccine-induced SARS-CoV-2 Seroprevalence JAMA 2021; 326(14): 1400-1409, which has as a co-author one of this manuscripts co-authors (MB).

Minor editorial comments:

- 1) For paragraph beginning on line 35, third sentence, suggest that the term “seroepidemiology” replacing “serology” would be more apt.
- 2) Line 106 \$ sign is missing to denote median household income.
- 3) Sentence, lines 182-184 ending – “... which may be learnt from in future measures.” Awkwardly written, suggest rewording.
- 4) References 8, 13, and 18 seem incomplete.

Reviewer #2 (Remarks to the Author):

Thank you for sharing this interesting and timely manuscript.

Summary

The paper presents a sero-epidemiology survey in San Francisco in February 2021, looking at differences in pathways to immunity against SARS-CoV-2 by race/ethnicity, age, and area of residence.

Residual blood samples from 1014 patients at two hospital networks from 4-17 February 2021 were analysed using two different assays to distinguish antibodies against SARS-CoV-2 resulting from infection (antibodies to the nucleocapsid), vs antibodies which could result from either infection or vaccination (antibodies to the spike protein). Bayesian statistics were used to estimate the probability of infection, and the probability of vaccination separately according to history of infection, according to age, ethnicity, and local area (zipcode).

The authors find evidence of inequality in vaccination uptake by race/ethnicity and local area suggesting inequalities by deprivation (illustrated with two examples). In addition, they find evidence suggesting a double inequity, in which population groups with highest probability of infection had the lowest probability of being vaccinated.

This is a timely and, to the best of my knowledge, novel contribution on the important topic of inequalities in COVID-19 vaccination and pathways to immunity. While this type of study is inevitably somewhat context-specific, the double inequity of lower vaccination of higher risk individuals is a phenomenon that's also been observed in other contexts (e.g Jain et al. PLoS One 2018 Nov 15; 13(11):e0207183 looking at shingles vaccination in the UK). It is likely to be generalisable to other high-income settings, and particularly in places with focus on public health than San Francisco.

Understanding inequalities in pathways to immunity is important for evaluating and improving fairness of vaccination programmes, and also for predicting future inequalities in health outcomes likely to result from long-term complications of COVID-19 infection rather than vaccination (such as long COVID).

General comments

1. The methods are clearly explained and provide sufficient detail for the work to be reproduced, and the authors have shared clearly annotated code on GitHub to support this.
2. The study appears to have some potential limitations which are not discussed – they are not necessarily major limitations, but it would be good to see them addressed in the discussion.
3. The implementation of the Bayesian statistical methods is outside my expertise. They are clearly explained to the general reader (Figure 1 in particular is very helpful) and are an appropriate choice of method to allow estimation of vaccination given infection status, but I would recommend statistical review.

Specific comments

Abstract

1. I found it difficult to work out what the methods were from this abstract. The two assays were only indirectly described and there was no mention of the Bayesian approach to estimation of vaccination given infection status. The methods were clearly described in the paper itself, but it would be helpful to compare the abstract against the WHO reporting guidance of sero-epidemiology studies and ensure the relevant elements such as the cross-sectional study design, laboratory methods (assays used), results with 95% CI/CrI, and key study limitations are included.

(World Health Organization Seroepidemiology Technical Working Group. ROSES-S: Statement from the World Health Organization on the reporting of seroepidemiologic studies for SARS-CoV-2, Influenza Other Respir Viruses. 2021 Sep; 15(5):561-568. doi: 10.1111/irv.12870).

2. Two point estimates are presented (infection risk for Latinx residents compared to White, vaccination of White residences over 65 compared to Black residents). The credible intervals for these should be included in the abstract.

3. The abstract refers to a finding that socio-economically-deprived zipcodes had higher infection probabilities and lower vaccination coverage than wealthier zipcodes. The manuscript provides two examples of zipcodes in San Francisco to illustrate this in the results (lines 100-108), but I did not find any systematic information on deprivation by area to support the general claim for a reader unfamiliar with San Francisco geography. Incorporating area deprivation as part of Figure 1 could address this.

Introduction

1. lines 24-28 Is there a reference available for the disparities in vaccine access and uptake, and for the role of barriers in access to healthcare, mistrust as a result of a history of medical racism and discrimination and poor access to reliable information about the vaccine?

Results

1. Line 83 – the authors describe a “slight” underrepresentation of individuals of Asian race/ethnicity, but proportion of people of Asian race/ethnicity is 20% lower than the proportion in the general population, so the ‘slight’ is debatable.

2. Lines 86-90 – the proportion of the study population with antibodies elicited by prior infection is low (8.6%). It might be helpful to provide some context for this result, e.g. infection rates in San Francisco in the first year of the pandemic to establish the plausibility of this result. There is relevant information in Supplementary Figure 4 but this does not appear to be referred to in the text of the manuscript.

Discussion

1. The discussion of potential limitations is brief. Some potential limitations the authors may consider addressing are:

a. Confounding by age

Age is stratified into 2 large groups (18-64 years and 65+ years) for analysis of inequalities. Could different age structures of the population groups by race/ethnicity and by area explain the associations observed (in part or whole)? The authors refer to an age-based roll-out of vaccination, and some information about the timing of vaccine rollout by age would be useful for understanding potential confounding by age with the two broad age categories used - for example, was everyone aged 18-64 eligible for the vaccine for the same length of time before the samples were taken 4-17 February, or was vaccine eligibility rolled out in age groups? Supplementary Figure 3 suggests that timing of vaccination rollout was similar for all adults 18-64 years, and could be used to address this potential limitation, but does not appear to be referred to in the manuscript.

b. Potential collider bias from selection based on healthcare use

The serosurveillance platform uses residual blood samples from individuals who have attended outpatient or inpatient services. Could the apparent inverse relationship between vaccination and infection result from collider bias by selecting on health care use? For example, if people with COVID-19 infection are more likely to attend health services than the general population, and also people who take up vaccination early are more likely to attend health services, there is a risk of finding an inverse relationship between infection and vaccination status. Potentially, this might differ by race/ethnicity or area (for example, if healthcare use is associated with proximity to health services). It seems unlikely to fully explain the large differences observed by race/ethnicity, but the potential issue of collider bias seems worth noting. Some context on the serosurveillance

platform might help with this (for example, who is eligible to use these hospital networks, what of samples are from inpatients vs outpatients?) might help the reader assess the potential for collider bias, and also the generalisability of the study.

c. Cross-sectional design

The analysis and interpretation assume that infection status predates vaccination (estimating vaccination status conditional on history of infection, and referring to lower vaccination among those with a higher probability of prior infection). If there has been time for vaccination to prevent infections among vaccinated individuals, then we would expect to see an inverse relationship between infection and vaccination. The introduction notes that the study captures the early roll-out of vaccination, and the proportion of people with antibodies is low, suggesting this is likely to be a reasonable assumption, but the limitations of the cross-sectional design should be mentioned, and a start date of the vaccine roll-out for over 65s and under 65s might also support the discussion. Supplementary Figure 3 could be used to support this discussion, but does not appear to be referred to in the manuscript.

d. Early roll out could be atypical

If this study captures only a very brief period at the start of the vaccine rollout, then it may describe inequalities in timeliness of vaccination over a short time frame, rather than inequalities in uptake over the course of the vaccine rollout. The recommendations for action may benefit from this caveat.

2. Lines 161-162 The authors note that "Like in San Francisco, the neighbourhoods which were predominantly Latinx communities and had higher indices of deprivation also had higher incidence of SARS-CoV-2 infection and lower vaccination coverage." However, without knowledge of San Francisco's geography and demographics to interpret the map in Figure 2, the race/ethnicity and deprivation characteristics of the neighbourhoods was not clear.

3. Lines 175-206 include a broad discussion on sources of structural inequalities, and recommendations to policy makers to address structural causes of health inequalities, which is important but the detail is perhaps not all directly relevant to this study. Compressing this might help address the potential limitations of the study in more depth. Alternatively, it might benefit from more use of references to support the arguments made, for example: lines 176-177 the relative investment in public health in San Francisco; line 178 (the role of incompatibility of living or work conditions with risk reduction in inequalities); lines 198-204 (recommendations for addressing inequalities).

Methods

1. Figure 1 is a helpful diagram for the general reader to understand the study outcomes being estimated, and could perhaps be usefully referred to in the methods section.

3. Line 300 – how many zipcodes were excluded due to including fewer than 10 individuals?

Tables and Figures

Table 1 please spell out the names of the hospital networks in the table or a footnote to it, and provide a reference for the American Community Survey in the table heading.

Figure 2 part a The boxplot is labelled as by hospital week, but the sample period only included 2 weeks – is this by day?

Figure 3 – would it be possible to include or incorporate a mapping of the % below the poverty line for this map, so that a reader unfamiliar with San Francisco can more readily see the association reported with deprivation? It would also be useful to include the location of the hospital services, if possible, to give some indication of proximity to health services.

Figure 5b is a data visualisation. The point estimates for the number of people vaccinated for every person naturally infected are presented by race/ethnicity among individuals aged 65+. For

people of Asian ethnicity the point estimate is 12.2 with a 95% CI 4.2-55.5. This is a wide CI and it is presented in very small font and not included in the graphic component of the data visualisation. Is it possible to include the CIs in the data visualisation graphic? Or if not, to make them more visible, perhaps with larger font?

Response to reviewers for “Using sero-epidemiology to monitor disparities in vaccination and infection with SARS-CoV-2”

We thank the reviewers for their thorough and helpful comments which have improved the manuscript considerably. Please see below for a point-by-point response to the reviewer’s comments.

Reviewer #1 (Remarks to the Author):

The authors describe seroepidemiologic study results of a convenience sample with about one thousand blood specimens from inpatients and outpatients collected at the University of California San Francisco Medical Center and the San Francisco Department of Public Health between February 4 and February 17, 2021, during the early stages of COVID-19 vaccination efforts. Serologic testing results for COVID-19 spike and nucleocapsid protein antibody were analyzed using binomial Bayesian modeling. The raw natural infection rate as measured by nucleocapsid antibody was 8.6% and spike protein IgG antibody measuring both natural and vaccine induced antibody was 28.4%. Model adjusted natural infection rates were highest in Hispanic and black residents, while the highest estimated vaccination rates were in Asian and white San Francisco study populations. Patients 65+ years had lower natural infection rates and as expected higher estimated vaccination rates given vaccination eligibility during that time period. Lower income medically underserved zip codes in San Francisco also had higher infection and lower estimated vaccination rates. The authors discuss the challenges and public health measures needed to address these sociodemographic disparities in COVID-19 risk and prevention.

This paper is well written and uses appropriate and strong laboratory and biostatistical methodology despite the study design weakness of patients’ blood specimens being a convenience sample (the authors address the relative representativeness of the study sample). Similar results regarding lower vaccination rates in areas with vulnerable populations have been seen in other parts of the United States but this rigorous seroepidemiologic analysis adds to the literature on the pandemic epidemiology in the U.S.

Specific comments:

1) Results, line 97 – The authors indicate that zip codes in southeastern San Francisco had the pattern of higher natural infection and lower vaccination rates across all age and demographic groups and reference Supplementary Table 2. Supplementary Table 2 does not stratify the zip code results by age and demographic groups. The low vaccination/high natural infection finding in these zip codes beg the question of whether these rates are related to the race/ethnic makeup of the zip codes or is an SES related phenomenon, or not an issue possible to disentangle.

This is an important point. Unfortunately stratifying results by both zipcode and ethnicity result in sample sizes which are underpowered to calculate the probability of vaccination or infection, which is why results are not reported stratified by both zipcode, race/ethnicity and age. However we have removed the reference to age and demography when reporting the zipcode results in line 95 for clarity. We also have added a new supplementary figure mapping race/ethnicity (Supplementary Figure 3a and 3b) and a metric of poverty by zipcode (Supplementary Figure 3c) to aid interpretation of the geographic disparities observed. Figure 3 a-c seems to indicate that, as the reviewer suggests, ethnicity/racial make-up of the zipcodes (potentially the underlying structural inequalities related to race/ethnicity) do seem to partially explain the results we observed, however unfortunately we are unable to estimate whether there is an independent zipcode effect.

2) Discussion paragraph, lines 175-179 – This one sentence paragraph is somewhat unclearly expressed. Structural inequalities in a wealthy progressive city such as San Francisco undoubtedly contribute to high infection rates and less vaccination in subpopulations. But does evidence exist for vaccine hesitancy, especially among the black population in San Francisco?

We agree that this paragraph is somewhat unclear so have restructured the paragraph to reflect this, and added citations to support claims made in lines 177-179. We have been unable to find peer reviewed studies on vaccine hesitancy in San Francisco's black population, however studies exploring vaccine hesitancy in other US contexts exist, and we have now referenced them when mentioning vaccine hesitancy in the introduction (lines 26-28). However given the lack of context specific evidence, we are cautious to state that hesitancy explains all or even most of the disparities in vaccination observed between black and white communities here when there is likely to be a complex interplay of factors contributing to the disparity we observe. We have added the phrase "and uptake of" to include hesitancy as a possible driver. We also have added a reference about initiatives specifically targeting the black community in the San Francisco Bay Area in the following paragraph.

3) Methods, lines 256-247 – The authors adjusted for manufacturer reported specificity for each serology assay. Manufacturer reported specificities are often overstated based on the set of samples used to assess specificity. The authors should perhaps comment on how lower test specificities would affect the uncertainty in the estimated rates of positivity.

A key issue we have worked on for SARS-CoV-2 serosurveillance is the impact of overstated sensitivities on seroprevalence estimation, which we have adjusted for in this analysis. We agree with the reviewer that overstated specificities would lead to over-estimating the true seroprevalences. As suggested, we have added a comment in the "Univariate data analysis (by assay)" section in the Methods about the need for unbiased test performance characteristics to yield unbiased estimates of seroprevalence.

4) Discussion, paragraph lines 181-195 – The authors reference a preprint describing an intervention in a predominantly Latinx community in San Francisco. Were there similar projects for the black population in San Francisco?

Yes, we welcome this point and have now added a reference to the Umoja Health initiative which was a community-academic partnership aimed at reducing COVID-19 health inequalities within the black community in the San Francisco Bay Area.

5) References – The authors should probably reference Jones J et al, Estimated US Infection- and Vaccine-induced SARS-CoV-2 Seroprevalence JAMA 2021;326(14):1400-1409, which has as a co-author one of this manuscript's co-authors (MB).

We have now added this reference.

Minor editorial comments:

1) For paragraph beginning on line 35, third sentence, suggest that the term “seroepidemiology” replacing “serology” would be more apt.

Done

2) Line 106 \$ sign is missing to denote median household income.

Done

3) Sentence, lines 182-184 ending – “... which may be learnt from in future measures.” Awkwardly written, suggest rewording.

Done

4) References 8, 13, and 18 seem incomplete.

Done - these references are reports and were reformatted incorrectly but have now been updated.

Reviewer #2 (Remarks to the Author):

Thank you for sharing this interesting and timely manuscript.

Summary

The paper presents a sero-epidemiology survey in San Francisco in February 2021, looking at differences in pathways to immunity against SARS-CoV-2 by race/ethnicity, age, and area of residence.

Residual blood samples from 1014 patients at two hospital networks from 4-17 February 2021 were analysed using two different assays to distinguish antibodies against SARS-CoV-2 resulting from infection (antibodies to the nucleocapsid), vs antibodies which could result from either infection or vaccination (antibodies to the spike protein). Bayesian statistics were used to estimate the probability of infection, and the probability

of vaccination separately according to history of infection, according to age, ethnicity, and local area (zipcode).

The authors find evidence of inequality in vaccination uptake by race/ethnicity and local area suggesting inequalities by deprivation (illustrated with two examples). In addition, they find evidence suggesting a double inequity, in which population groups with highest probability of infection had the lowest probability of being vaccinated.

This is a timely and, to the best of my knowledge, novel contribution on the important topic of inequalities in COVID-19 vaccination and pathways to immunity. While this type of study is inevitably somewhat context-specific, the double inequity of lower vaccination of higher risk individuals is a phenomenon that's also been observed in other contexts (e.g Jain et al. PLoS One 2018 Nov 15;13(11):e0207183 looking at shingles vaccination in the UK). It is likely to be generalisable to other high-income settings, and particularly in places with focus on public health than San Francisco.

Understanding inequalities in pathways to immunity is important for evaluating and improving fairness of vaccination programmes, and also for predicting future inequalities in health outcomes likely to result from long-term complications of COVID-19 infection rather than vaccination (such as long COVID).

General comments

1. The methods are clearly explained and provide sufficient detail for the work to be reproduced, and the authors have shared clearly annotated code on GitHub to support this.
2. The study appears to have some potential limitations which are not discussed – they are not necessarily major limitations, but it would be good to see them addressed in the discussion.

We appreciate the reviewer's thoughtful comments around the need to address additional limitations. We have attempted to address these, as detailed below.

3. The implementation of the Bayesian statistical methods is outside my expertise. They are clearly explained to the general reader (Figure 1 in particular is very helpful) and are an appropriate choice of method to allow estimation of vaccination given infection status, but I would recommend statistical review.

We agree that Figure 1 is a useful schematic, and have included a reference to it in the "Bivariate data analysis (infection vs. vaccination)" section of the Methods.

Specific comments

Abstract

1. I found it difficult to work out what the methods were from this abstract. The two assays were only indirectly described and there was no mention of the Bayesian approach to estimation of vaccination given infection status. The methods were clearly described in the paper itself, but it would be helpful to compare the abstract against the WHO reporting guidance of sero-epidemiology studies and ensure the relevant elements such as the cross-sectional study design, laboratory methods (assays used), results with 95% CI/CrI, and key study limitations are included.

(World Health Organization Seroepidemiology Technical Working Group. ROSES-S: Statement from the World Health Organization on the reporting of seroepidemiologic studies for SARS-CoV-2. *Influenza Other Respir Viruses*. 2021 Sep;15(5):561-568. doi: 10.1111/irv.12870).

We thank the reviewer for pointing this out – we have updated the abstract to include the relevant elements as suggested.

2. Two point estimates are presented (infection risk for Latinx residents compared to White, vaccination of White residences over 65 compared to Black residents). The credible intervals for these should be included in the abstract.

We have now updated the abstract to include the credible intervals.

3. The abstract refers to a finding that socio-economically-deprived zipcodes had higher infection probabilities and lower vaccination coverage than wealthier zipcodes. The manuscript provides two examples of zipcodes in San Francisco to illustrate this in the results (lines 100-108), but I did not find any systematic information on deprivation by area to support the general claim for a reader unfamiliar with San Francisco geography. Incorporating area deprivation as part of Figure 1 could address this.

We have now included a new supplementary figure to address this which incorporates a metric of poverty mapped by zipcode (Supplementary Figure 3c).

Introduction

1. lines 24-28 Is there a reference available for the disparities in vaccine access and uptake, and for the role of barriers in access to healthcare, mistrust as a result of a history of medical racism and discrimination and poor access to reliable information about the vaccine?

Thanks for this helpful suggestion. We have now added appropriate references for all the mentioned disparities in this section.

Results

1. Line 83 – the authors describe a “slight” underrepresentation of individuals of Asian race/ethnicity, but proportion of people of Asian race/ethnicity is 20% lower than the proportion in the general population, so the ‘slight’ is debatable.

We have now removed the word “slight”.

2. Lines 86-90 – the proportion of the study population with antibodies elicited by prior infection is low (8.6%). It might be helpful to provide some context for this result, e.g. infection rates in San Francisco in the first year of the pandemic to establish the plausibility of this result. There is relevant information in Supplementary Figure 4 but this does not appear to be referred to in the text of the manuscript.

We have included a reference to Supplementary Figure 4 (Now re-numbered to Supplementary Figure 5) in the methods section.

Discussion

1. The discussion of potential limitations is brief. Some potential limitations the authors may consider addressing are:

We thank the reviewer for these helpful points below, and have expanded the section on caveats and limitations in the Discussion section in line with their suggestions.

a. Confounding by age

Age is stratified into 2 large groups (18-64 years and 65+ years) for analysis of inequalities. Could different age structures of the population groups by race/ethnicity and by area explain the associations observed (in part or whole)? The authors refer to an age-based roll-out of vaccination, and some information about the timing of vaccine rollout by age would be useful for understanding potential confounding by age with the two broad age categories used - for example, was everyone aged 18-64 eligible for the vaccine for the same length of time before the samples were taken 4-17 February, or was vaccine eligibility rolled out in age groups? Supplementary Figure 3 suggests that timing of vaccination rollout was similar for all adults 18-64 years, and could be used to address this potential limitation, but does not appear to be referred to in the manuscript.

We have provided some additional context on vaccine roll-out in San Francisco to justify our decision to use these broad age groups. Eligibility for vaccination during the time of our sampling (Feb 4-17, 2021) was based either by age (65 yr or older) or, for individuals under 65 years, by profession (healthcare workers). We have also added a reference to Supplementary Figure 3 (now renumbered to Supplementary Figure 4) to highlight the timing of vaccination by age.

b. Potential collider bias from selection based on healthcare use

The serosurveillance platform uses residual blood samples from individuals who have attended outpatient or inpatient services. Could the apparent inverse relationship between vaccination and infection result from collider bias by selecting on health care use? For example, if people with COVID-19 infection are more likely to attend health services than the general population, and also people who take up vaccination early are more likely to attend health services, there is a risk of finding an inverse relationship between infection and vaccination status. Potentially, this might differ by race/ethnicity or area (for example, if healthcare use is associated with proximity to health services). It seems unlikely to fully explain the large differences observed by race/ethnicity, but the potential issue of collider bias seems worth noting. Some context on the serosurveillance platform might help with this (for example, who is eligible to use these hospital networks, what of samples are from inpatients vs outpatients?) might help the reader assess the potential for collider bias, and also the generalisability of the study.

The exclusion and inclusion criteria for this study were outlined very thoroughly in the initial paper arising from this serosurvey, and we now realise that we did not include sufficient detail in this manuscript for the reader to assess the potential for this type of bias without referring to this other paper.

We designed our inclusion and exclusion criteria to mitigate potential selection biases, including several of the concerns brought up by the reviewer. Firstly, the only inpatients who were included in this study were under 18s and non-COVID emergency room visits - we did not include any other inpatients in the study. We also excluded any visits where the patient was visiting for COVID-19 symptoms or under suspicion of COVID, and also only selected samples which were collected from Complete Metabolic Panels, which is a routine blood test. These criteria reduce the chance of capturing chronically ill individuals, or individuals who are more susceptible to symptomatic/severe COVID. In addition, we selected samples from both the UCSF health network and the San Francisco Department of Public Health network, which serves the under-insured and uninsured population, including San Francisco's homeless population, meaning we captured individuals who may not always be captured by surveillance systems as they have interactions with the city's public health department rather than traditional healthcare providers. This is now more clearly explained in the Data sub-section of the Methods section, and we have expanded the limitations section in the discussion to more explicitly discuss these issues.

c. Cross-sectional design

The analysis and interpretation assume that infection status predates vaccination (estimating vaccination status conditional on history of infection, and referring to lower vaccination among those with a higher probability of prior infection). If there has been time for vaccination to prevent infections among vaccinated individuals, then we would expect to see an inverse relationship between infection and vaccination. The introduction notes that the study captures the early roll-out of vaccination, and the

proportion of people with antibodies is low, suggesting this is likely to be a reasonable assumption, but the limitations of the cross-sectional design should be mentioned, and a start date of the vaccine roll-out for over 65s and under 65s might also support the discussion. Supplementary Figure 3 could be used to support this discussion, but does not appear to be referred to in the manuscript.

We realized that our use of the term “prior infection” is confusing, since we are using it to refer to “infection prior to the serosurvey” and not “infection prior to being vaccinated”. We agree with the reviewer that given the cross-sectional study, we are not able to determine the relative timing of infection and vaccination within an individual, and this is a limitation -- however, this was not the intention of the work, and we have made this clarification when we first describe measuring “prior infection”. We note that given the short duration between the start of vaccine roll-out in San Francisco and this serosurvey, we expect that most of the infections that occurred prior to the serosurvey among individuals also occurred prior to when they were vaccinated.

d. Early roll out could be atypical

If this study captures only a very brief period at the start of the vaccine rollout, then it may describe inequalities in timeliness of vaccination over a short time frame, rather than inequalities in uptake over the course of the vaccine rollout. The recommendations for action may benefit from this caveat.

Given the short duration between the start of vaccine roll-out in San Francisco and this serosurvey, we agree that our data is only able to capture vaccination coverage and inequities therein during the early roll-out. We have underscored this caveat as well.

2. Lines 161-162 The authors note that “Like in San Francisco, the neighbourhoods which were predominantly Latinx communities and had higher indices of deprivation also had higher incidence of SARS-CoV-2 infection and lower vaccination coverage.” However, without knowledge of San Francisco’s geography and demographics to interpret the map in Figure 2, the race/ethnicity and deprivation characteristics of the neighbourhoods was not clear.

We have now made this clearer, please see supplementary Figure 3a-c which plots a) the proportion of the population identifying as Hispanic/Latinx by zipcode b) the proportion of the population identifying as Black/African American by zipcode and c) the proportion of the population with a household income below the poverty threshold. All data come from the American Community Survey 2019 5 year estimates, cited in the manuscript.

3. Lines 175-206 include a broad discussion on sources of structural inequalities, and recommendations to policy makers to address structural causes of health inequalities, which is important but the detail is perhaps not all directly relevant to this study. Compressing this might help address the potential limitations of the study in more

depth. Alternatively, it might benefit from more use of references to support the arguments made, for example: lines 176-177 the relative investment in public health in San Francisco; line 178 (the role of incompatibility of living or work conditions with risk reduction in inequalities); lines 198-204 (recommendations for addressing inequalities).

We have now added additional references in lines 175-206 to support all of the examples mentioned by the reviewer.

Methods

1. Figure 1 is a helpful diagram for the general reader to understand the study outcomes being estimated, and could perhaps be usefully referred to in the methods section.

We agree and now include a reference to Figure 1 in the methods section.

2. Line 300 – how many zipcodes were excluded due to including fewer than 10 individuals?

13 zipcodes were excluded. These zipcodes cover a small area of downtown San Francisco (94104, 94111, 94119, 94126), a large city park (94129), an island (94130) or are unusual zipcodes which cover very small areas of the city (94142, 94143, 94145, 94146, 94147, 94159), and as can be seen from Figure 2b, the distribution of the sample population by zipcode is proportional to the population of San Francisco residing in each zipcode.

Tables and Figures

Table 1 please spell out the names of the hospital networks in the table or a footnote to it, and provide a reference for the American Community Survey in the table heading.

Done as suggested.

Figure 2 part a The boxplot is labelled as by hospital week, but the sample period only included 2 weeks – is this by day?

Figure caption was incorrect - the results are just by hospital covering the full two weeks of the study. This is now corrected.

Figure 3 – would it be possible to include or incorporate a mapping of the % below the poverty line for this map, so that a reader unfamiliar with San Francisco can more readily see the association reported with deprivation? It would also be useful to include the location of the hospital services, if possible, to give some indication of proximity to health services.

A new figure showing the proportion of the population below the poverty line has been added to as a new supplementary figure (5c). However the San Francisco Department of Public Health operates throughout the city, and the UCSF Health Network operates across a network of many locations <https://campusplanning.ucsf.edu/campus/other-ucsf-locations> which may reduce the utility of plotting hospital locations.

Figure 5b is a data visualisation. The point estimates for the number of people vaccinated for every person naturally infected are presented by race/ethnicity among individuals aged 65+. For people of Asian ethnicity the point estimate is 12.2 with a 95% CI 4.2-55.5. This is a wide CI and it is presented in very small font and not included in the graphic component of the data visualisation. Is it possible to include the CIs in the data visualisation graphic? Or if not, to make them more visible, perhaps with larger font?

Thank you for this comment, we have now increased the font size for the CIs.

REVIEWERS' COMMENTS

Reviewer #1 (Remarks to the Author):

The author have responded to reviewer comments in a thorough and thoughtful manner.

Specific comments:

1) Discussion, line 162 - Minor editorial suggestion to change sentence beginning to "As in San Francisco,..."

2) This reviewer is slightly confused by Supplementary Figure 4 which seems to indicate a small number of 0-17 were vaccinated in San Francisco by February 2021 whereas the manuscript (lines 296-298) states this age group was not eligible for vaccination at that time.

Reviewer #2 (Remarks to the Author):

The authors have fully addressed my comments - thank you.

REVIEWERS' COMMENTS

Reviewer #1 (Remarks to the Author):

The author have responded to reviewer comments in a thorough and thoughtful manner.

Specific comments:

1) Discussion, line 162 - Minor editorial suggestion to change sentence beginning to "As in San Francisco,..."

Thank you, this change has been made

2) This reviewer is slightly confused by Supplementary Figure 4 which seems to indicate a small number of 0-17 were vaccinated in San Francisco by February 2021 whereas the manuscript (lines 296-298)states this age group was not eligible for vaccination at that time.

We thank the reviewer for pointing this out and agree that this is confusing. These data come from San Francisco Department of Public Health. San Francisco was in the 1A phase of vaccination (which only included healthcare workers, workers in long-term care facilities and over 65s) until February 24. Note this is a log scale, so the numbers of under 18s vaccinated are very small (approximately 0.1% of under 18 year olds). We have double checked these data, which are publicly available, and they are consistent with Supplementary Figure 4. We cannot determine whether this was the result of vaccinations being given early despite ineligibility or whether there were errors in reporting age for a small number of vaccination records.

Reviewer #2 (Remarks to the Author):

The authors have fully addressed my comments - thank you.

Thanks for the helpful comments